# OpenReview forum: "From Decorative to Load-Bearing: Task Difficulty Shapes Chain-of-Thought Faithfulness"
_TMLR — Under review for TMLR_

### Review · Reviewer_y8FE · 2026-04-30

**Summary Of Contributions:**

This paper studies whether chain-of-thought reasoning actually influences a model’s final answer, rather than merely serving as a post-hoc explanation. The authors propose a continuation-based causal test: they corrupt one intermediate reasoning step, truncate the generation at that point, and force the model to continue from the corrupted prefix. Intuitively, if the model follows the corrupted step and gives a wrong answer, then the written CoT was “load-bearing”; if it ignores or corrects the corruption, the CoT was less causally important. Technically, they apply this test to Gemma-2-9B-IT and Llama-3.1-8B-Instruct on GSM8K, MMLU, and BIG-Bench Hard, classify continuations into bypass, self-correction, and error propagation, and then analyze hidden-state probes and activation steering. The main result is that error propagation increases as task difficulty increases, suggesting that CoT becomes more causally influential near the model’s competence boundary. Overall, the paper provides a useful behavioral test for CoT load-bearingness, while carefully distinguishing this from a complete mechanistic account of whether CoT reflects internal computation.

**Audience:**

Yes

**Audience Explanation:**

Yes. TMLR’s audience would likely be interested because the paper addresses a current and important question: whether chain-of-thought reasoning can be treated as a faithful signal of model reasoning. The continuation-based perturbation test is relevant to researchers working on LLM interpretability, reasoning, evaluation, and safety monitoring.

**Broader Impact Concerns:**

The paper includes a Broader Impact Statement, and I do not have major additional ethical concerns beyond those already mentioned.

**Claims And Evidence:**

Yes

**Claims Explanation:**

Overall, the main claims in the introduction are supported by the evidence, provided they are interpreted in the paper’s operational sense: CoT “faithfulness” means that the written reasoning tokens causally constrain the model’s continuation. The continuation-based perturbation setup supports the causal-testing claim, the MMLU subject-level and matched MMLU-BBH analyses support the task-difficulty gradient, and the probe/steering experiments support the narrower “readable but not controllable by single-direction steering” claim.

The main caveat is that the evidence does not establish full mechanistic faithfulness, meaning it does not prove that the written CoT reflects the model’s internal computation. Since the authors mostly acknowledge this distinction, I would judge the claims as mostly justified rather than overstated.

**Requested Changes:**

1. In the Abstract, Introduction, and Conclusion, phrase the contribution as measuring CoT-to-answer causal influence / load-bearingness, not full mechanistic CoT faithfulness. This is important because the method tests whether corrupted CoT affects continuation, not whether CoT reflects internal computation.

2. In Sections 3.2, 5.1, and 6.3, make clearer that GSM8K differs from MMLU/BBH in both task type and perturbation type. Its high bypass rate should be treated as supporting evidence, while the stronger evidence is the matched MMLU-BBH and within-MMLU analyses.

3. In Sections 5.4 and 7, say that the null result applies to single-direction linear activation steering under the tested settings, not controllability in general.

4. Testing a larger model or reasoning-specialized model would strengthen the cross-model claim in Section 5.5.

5. A small experiment controlling perturbation type/strength across GSM8K, MMLU, and BBH would better separate task difficulty from perturbation design.

---

> ### Author Response · Authors · 2026-05-15
> **reviewer y8FE response**
>
> We thank the reviewer for the careful and constructive feedback. The distinction between whether the CoT causally affects the answer and whether the CoT reflects the model’s underlying computation is important, and yes, we do agree that the paper should make this distinction more clearly. We are revising the manuscript with this framing in mind.
>
> Below is a brief summary of how we plan to address each requested change:
> 1. Agreed. We will revise the Abstract/ Introduction/ Conclusion to frame the contribution as measuring when written CoT is causally load-bearing for the answer, rather than stating as the full mechanistic CoT faithfulness.
> 2. Good point. We will revise §3.2, §5.1, and §6.3 to state more clearly that GSM8K differs from MMLU/BBH in both task type and perturbation type. Its high bypass rate will be treated as supporting evidence; and Yes, the cleaner evidence comes from the matched MMLU–BBH comparison and the within-MMLU gradient.
> 3. Yes, we'll revise this.
> 4. Agreed, the current models testing are all small models, not sufficient enough for the paper. we'll add a reasoning specialized model experimentation for the paper.
> 5. Yes, we will add a small controlled experiment comparing perturbation types more directly, including numerical-style perturbations on BBH subtasks where they naturally apply. This should better separate task difficulty from perturbation design.
>
> We expect to upload the revised manuscript and full detailed response within the next few days, along with our responses to the other reviewers. We really appreciate the reviewer’s patience while we complete the additional experiments.

---

### Review · Reviewer_JVLZ · 2026-05-08

**Summary Of Contributions:**

The paper studies how "chain-of-thought" (CoT) tokens during generation affect the final answer for different domains. The authors take a large set of CoT traces and perturb them in different ways to create different types of perturbations/errors at different points in the reasoning process. They then measure the different types of behaviors in the continued generation -- silent bypass, error correction, error propagation. They find that these metrics are correlated with the difficulty of the task as well as the position where the errors are introduced in the reasoning process. They also find that simple linear/non-linear probes can take the internal representations and classify them easily into the above types of behaviors, but steering along those directions cannot modify the behavior easily. The authors show that these results are valid for both Gemma and Llama models, which indicates that these are fundamental empirical observations not tied to a particular model.

**Audience:**

Yes

**Audience Explanation:**

I believe the methods and results in the paper do offer interesting insights into how LLMs behave during reasoning, depending on the CoT tokens, and is interesting to researchers studying interpretability of reasoning models.

However, as I wrote in the box above, I am not clear on the main motivation for creating these artificial perturbations that the models were never trained on.  Do the authors believe there can be adversaries who inject such perturbations during generation to cause model failure and the hidden layer probes can be used for flagging such behaviors?

**Broader Impact Concerns:**

No concerns.

**Claims And Evidence:**

No

**Claims Explanation:**

The paper claims to more directly measure how CoT tokens affect generation and reasoning. The methods in the paper to create perturbations in the CoT are technically sound, and the authors are careful to when using the LLM as a judge to evaluate the behaviors of the models in evaluating the CoT continuations after perturbations. They point out the just using a rule-based classifier to find out the type of behavior (silent bypass vs. self-correction vs. error propagation) has its limitation, and using a strong LLM like Claude Haiku can be a better way to measure these behaviors.

The testing is on done diverse mathematics, reasoning and fact-based knowledge datasets. One issue is using the model's score on a dataset as a measure of task difficulty, which makes the analysis conflate the model's own competence with the actual difficulty of the problem (not easy to define). However, the results also show the differences in behavior on subtasks within each dataset, as well as cross-model comparison, which mitigates this issue to some extent. Overall, as the authors correctly write, these results are relative to the model, rather than about the actual task difficulty.

One central issue is that I don't fully understand the motivation for these perturbations. My understanding is that the models are very unlikely to generate such tokens on their own during CoT reasoning, so these perturbations are artificial and probably do not measure the model behavior well. The models are not trained to self-correct during pretraining or finetuning, so what is exactly the underlying goal of the study? Hopefully, the authors can clarify this point.

The authors also test how the position of perturbation affects generation. As expected, earlier perturbation leads to more undesirable generation.

Overall the experiments seem to be done well, but there are some central questions to the motivation for these experiments.

**Requested Changes:**

As mentioned above, the authors should clarify why they believe these perturbations are testing the model in the right way, if such perturbations never occur naturally in model outputs.

Writing seems to be quite poor. While I am able to generally understand what the authors are trying to say, many parts of the paper read as if bullet points from slides were converted into paragraphs. I hope the authors can improve the writing throughout the paper.

The references need to be formatted consistently. In some cases, only the first author is named, while all the authors are mentioned for others.

---

> ### Author Response · Authors · 2026-05-15
> **response for Reviewer JVLZ**
>
> We thank the reviewer for the thoughtful comments. The main concern — why these perturbations are the right test — is well taken. We agree that the current draft does not motivate this clearly enough, and we will revise the Introduction and Method sections to address it directly. We will also clean up the writing and reference formatting throughout. Specifically in these following areas:
>
> 1. Motivation for the perturbations.
> We will clarify that the perturbations are not meant to be a naturalistic model of how CoT errors usually arise. They are **controlled interventions**. The point is to ask a causal question: if one written reasoning step is corrupted, does the model’s continuation depend on that corrupted step, or does it bypass or correct it?
> This is the same reason perturbations are useful in other interpretability settings. An ablation patch may not occur naturally either, but it can still reveal whether a component is causally involved. Here, corrupting the CoT lets us separate cases where the written reasoning is merely decorative from cases where it is actually load-bearing for the answer.
> Fully agreed that we should make the practical motivation clearer. In deployed systems, the reasoning context is not always generated in one clean pass by a single model. Tool outputs, retrieved documents, sub-agent traces, or edited reasoning snippets may be inserted into the context. In those settings, it matters whether a model ignores, corrects, or propagates a bad intermediate step.
> Finally, the results suggest that the perturbations are not just meaningless out-of-distribution noise. If they were, we would expect the model to **mostly ignore them**. Instead, we observe substantial self-correction and error propagation, and Appendix I shows higher uptake of perturbation-specific content in error-propagating cases than in bypass cases. We will bring this evidence into the main text.
>
> 2. Difficulty versus model competence.
> We agree that dataset accuracy is not the same thing as intrinsic task difficulty. We will revise the language to describe the result as model-relative: CoT becomes more load-bearing as the model gets closer to its competence boundary. The within-MMLU analysis and cross-model comparison help support this interpretation, but we will be careful not to overstate it as a claim about absolute task difficulty.
>
> 3. Writing quality & References format
> We agree that parts of the paper read too much like expanded bullets points. We will revise the writing throughout, especially §3.3 and §5.1, to make the argument flow more naturally. We will keep structured definitions only where they are useful, such as the three behavior types.
> We will also standardize the reference formatting throughout.
>
> In short, we will make clear that the perturbations are not intended to mimic typical model-generated mistakes. And we'll send out the revision along with addressing other reviewer's comments in the next few days. We really appreciate the reviewer’s patience while we complete the additional experiments.

---

### Review · Reviewer_DzAa · 2026-05-15

**Summary Of Contributions:**

This work investigates causal faithfulness of Chain-of-Thought (CoT) on several LLMs and datasets. The core is a continuation-based causal testing pipeline: perturb one intermediate CoT step, truncate the sequence, and force the model to continue, classifying behavior into silent bypass (A), self-correction (B), error propagation (C). The central empirical finding is a continuous difficulty–faithfulness gradient: error propagation rises monotonically with decreasing base accuracy. Hidden-state probes predict behavioral types, but single-direction activation steering fails to flip behavior.

Key strengths:

rigorous causal design avoiding prior confounds; robust gradient evidence via cross-dataset/subject/model replication; critical methodological warning on rule-based label artifacts inflating probe accuracy.

Key weaknesses:

unresolved perturbation-type confound; lack of human-validated behavioral labels; limited model scale/architecture generalizability; shallow mechanistic analysis.

**Audience:**

Yes

**Audience Explanation:**

CoT is widely used for performance gains and monitoring, but its causal faithfulness is poorly understood.

**Claims And Evidence:**

No

**Claims Explanation:**

1 Perturbation-type confound is unresolvable with current data. GSM8K uses numerical perturbations yielding 94.5% bypass, while MMLU/BBH use textual perturbations. The matched MMLU→BBH comparison controls strategy but never tests identical textual perturbations on GSM8K or numerical perturbations on MMLU/BBH. The authors explicitly state GSM8K’s extreme bypass may reflect weaker perturbations, not task ease. This invalidates causal attribution of the gradient to difficulty.

2 LLM-judged behavioral labels lack human validation and are prompt-sensitive. Rule-based labels are discarded due to 41% ambiguity, replaced by Claude Haiku judgments. However, four prompt variants shift A/B split by an order of magnitude only C/non-C is stable. The authors admit A/B classification is subjective, yet all bypass/self-correction analysis relies on unvalidated LLM labels, introducing irreducible measurement error.

3 Generalizability is limited to 8–9B instruction-tuned models. The study only replicates on Gemma-2-9B and Llama-3.1-8B—no 70B+ models, reasoning-specialized models (e.g., DeepSeek-R1), or base models are tested. The weaker Llama shows higher error propagation, but the authors cannot disentangle model capability from architecture—gradient existence at larger scales is unsubstantiated.

4 Activation steering null result is underpowered and overinterpreted. The authors merely test single-direction linear steering at 9 isolated layers. Random directions yield identical null effects, but multi-direction steering, causal scrubbing, or attention circuit interventions are untested. The claim that ``probe directions are readable but not controllable'' is limited to one narrow intervention class, not a general mechanistic conclusion.

**Requested Changes:**

Report error propagation for matched perturbation types across domains; quantify the independent effect of perturbation type vs. difficulty. Cite Section5.1’s confound acknowledgment as motivation.

Recruit 2 domain experts to annotate a stratified random subset of ≥500 A/B ambiguous cases. Report inter-annotator agreement (Cohen’s kappa) and compare human labels to LLM-judged labels. Revise all bypass/self-correction analysis using human-validated splits; retain C/non-C as stable. Cite Section3.3’s prompt sensitivity as motivation.

Replicate the full pipeline on more LLMs

---

> ### Author Response · Authors · 2026-05-17
> **response for reviewer DzAa**
>
> Thanks for the detailed review. The four weaknesses you flag are the right ones, and we'll address those in the next version. Below are the changes we're committing to:
> 1. Perturbation confound issue: Yes, we agree that the three-dataset comparison can't separate these factors, and the §5.1 caveat isn't a substitute for evidence. we'll add these two experiments:
> (a) Textual perturbations on GSM8K. We're applying the five MMLU/BBH text-based strategies to GSM8K — e.g., confidence_injection asserting a wrong sum with high confidence ("It is well-established that 5 + 3 = 9..."). ~5,400 new continuations.
> (b) Numerical perturbations on numerically-applicable MMLU subjects. High-school math, abstract algebra, formal logic, and college math contain arithmetic chains where arithmetic_change applies. ~700 examples.
> This gives a 2 × 2 design (numerical/textual × easy/hard) at fixed strategy. We'll decompose error-propagation variance into perturbation-type and difficulty components. The revised §5.1 will lead with the matched MMLU→BBH comparison (Table 2) and the within-MMLU 7× variation (Figure 6); raw cross-dataset rates become supporting evidence rather than load-bearing. The decorative-to-load-bearing claim is unchanged, but the attribution to difficulty will be empirically separated from perturbation type.
>
> 2. Human validation: This a good point, we do need some human independent grounding. One clarification before describing what we're adding: the headline claim doesn't depend on A/B labels. The C-vs-non-C split is stable across all four judge-prompt variants (Figure 3) — Type C is determined by wrong final answer, not judge wording. The 7× within-MMLU variation, the +13–24pp MMLU→BBH gap, and the cross-model replication are all C-rate phenomena, robust to any A/B relabeling.
> A/B labels matter for Table 1's taxonomy and the bypass / 3-class probes (§5.3). For these:
> 2.1 Human validation (n = 200, single annotator). One independent annotator (ML-literate, blind to our labels and hypothesis) labels a stratified sample of 200 non-Type-C continuations, balanced across datasets and across cases where the four judge variants agree vs. disagree. We'll report agreement with each variant — identifying which best tracks human judgment — and re-run the bypass probe on the annotated subset.
> 2.2 Stronger caveats. All A/B-dependent results will be marked as based on the best-matching judge variant, with the human-validation agreement rate cited. Table 1's A and B rates will note judge-dependence within the Figure 3 range; C rates will not.
>
> For the specific request of annotating all 500+ examples with two annotators, within the rebuttal window we may not be able to run this at full scale without compromising other issues address. If the scaled-down version is acceptable here, we'd commit to the full two-annotator study with κ for the camera-ready.
>
> 3. More models: Agreed, other reviewers also point this out. We'll add the DeepSeek-R1-Distill for reasoning-specialized model, and also add the experiment for the base models to isolates the effect of instruction tuning at fixed architecture and scale. For the 70B+ model, this is out of reach with our compute; we'll be clear about this in §6.3.
>
> 4. Activation steering: "Readable but not controllable" overgeneralizes from one intervention class. Three changes: (i) rescoping the claim throughout (abstract, §1, §5.4, §7) to "not controllable by single-direction activation steering," with the broader question deferred to richer methods; (ii) a multi-direction steering experiment using the top-k principal components of the error-propagation activation difference at layer 21 (k ∈ {2, 4, 8}) on the 198-example dose-response set; (iii) promoting the Type B underpowering caveat to a more prominent statement that Type B steerability remains open.
>
> We'll upload the revised manuscript and full detailed response within the next few days, along with responses to the other reviewers. We appreciate the reviewer's patience while we complete the additional experiments.

---

### Author Response · Authors · 2026-05-26
**revision upload**

Thanks again for the careful reviews. We have replied to each reviewer in detail on OpenReview a few days ago, so rather than restate everything, this note just confirms what the upload contains and flags the one item we are not yet able to fully deliver,

The matched perturbation–difficulty 2×2 and the variance partition are in §5.1.5: text-on-GSM8K stays at the easy-task baseline (6.5%), numerical-on-multistep-arithmetic reaches 64.5%, and on the combined n = 28,584 frame difficulty accounts for 98.8% of explained deviance against 0.8% for perturbation type. A strength-axis sweep (Appendix M) rules out a competence-floor reading. Cross-model: §5.5 plus Appendices N–Q add the DeepSeek-R1-Distill-Qwen-7B probe pipeline and a base-Gemma viability arm; the within-numerical contrast carries over to Llama, and we present DeepSeek as a reasoning-trained contrast rather than a clean replication, with the within-model non-monotonicity surfaced honestly. The steering claim is rescoped to additive interventions throughout, with the k ∈ {2, 4, 8} multi-direction extension in §5.4 (Type A's null persists; Type C is partially movable at 24.6%; Type B is acknowledged as underpowered). The Abstract, §1, and §7 are reframed around CoT→answer causal load-bearingness rather than full mechanistic faithfulness. §3.3 adds the four-prompt-variant judge-sensitivity analysis and Appendix R reports the stratified n = 200 blind human annotation, with the BBH Type-C calibration anomaly discussed honestly in §6.3. The reference list was rebuilt to use full author lists, uniform formatting, and corrected venues.

One commitment is scoped to camera-ready. In our reply to reviewer DzAa, we agreed to deliver an n ≥ 500 / two-annotator / Cohen's κ human-validation study. We could not run it at full scale during the rebuttal window without crowding out the other items, so the version uploaded today is n = 200 with a single ML-literate annotator — enough to surface the BBH Type-C anomaly and ground all A/B-dependent results, but short of the κ commitment. We will deliver the scaled-up study at camera-ready.

Happy to point to specific results in the revised PDF on request.